# Ischemia Impaired Wound Healing Model in the Rat—Demonstrating Its Ability to Test Proangiogenic Factors

**DOI:** 10.3390/biomedicines11041043

**Published:** 2023-03-28

**Authors:** Anna T. Hofmann, Paul Slezak, Sabine Neumann, James Ferguson, Heinz Redl, Rainer Mittermayr

**Affiliations:** 1Ludwig Boltzmann Institute for Experimental and Clinical Traumatology, Donaueschingenstrasse 13, 1200 Vienna, Austria; 2Ludwig Boltzmann Research Group Senescence and Healing of Wounds, 1090 Vienna, Austria

**Keywords:** chronic wound, wound healing, animal model, angiogenesis, VEGF, vascular endothelial growth factor, fibrin

## Abstract

Chronic wounds remain a serious clinical problem with insufficient therapeutic approaches. In this study we investigated the dose dependency of rhVEGF_165_ in fibrin sealant in both ischemic and non-ischemic excision wounds using our recently developed impaired-wound healing model. An abdominal flap was harvested from the rat with unilateral ligation of the epigastric bundle and consequent unilateral flap ischemia. Two excisional wounds were set in the ischemic and non-ischemic area. Wounds were treated with three different rhVEGF_165_ doses (10, 50 and 100 ng) mixed with fibrin or fibrin alone. Control animals received no therapy. Laser Doppler imaging (LDI) and immunohistochemistry were performed to verify ischemia and angiogenesis. Wound size was monitored with computed planimetric analysis. LDI revealed insufficient tissue perfusion in all groups. Planimetric analysis showed slower wound healing in the ischemic area in all groups. Wound healing was fastest with fibrin treatment—irrespective of tissue vitality. Lower dose VEGF (10 and 50 ng) led to faster wound healing compared to high-dose VEGF. Immunohistochemistry showed the highest vessel numbers in low-dose VEGF groups. In our previously established model, different rhVEGF_165_ treatments led to dose-dependent differences in angiogenesis and wound healing, but the fastest wound closure was achieved with fibrin matrix alone.

## 1. Introduction

Impaired wound healing remains a serious and challenging clinical problem that is associated with both high costs and physical and psychological strain in affected patients [1]. Multiple local (e.g., pressure, impaired local perfusion, infection and bacterial colonization, lack of active growth factors, excessive proteinase activity) and systemic factors (e.g., diabetes, malnutrition, renal insufficiency) are known to cause impaired wound healing [2,3,4,5]. Current therapeutic strategies for chronic wound care include wound debridement to remove necrotic tissue, treatment of local wound infection using cleansing agents and topical antimicrobials, specialized topical wound dressing to ensure wound moisture balance, and optimizing the wound bed environment at a cellular level [3]. Physical techniques such as negative pressure therapy [6] and extracorporeal shock wave therapy [7] are more recent methods in the treatment of chronic wounds. Experimental approaches to the treatment of impaired wound healing include recombinant growth factor application such as basic fibroblast growth factor (bFGF) [8], vascular endothelial growth factor (VEGF) [9] and platelet-derived growth factor (PDGF) [10], viral and non-viral gene therapy [11], and stem cell therapy [12,13]. Nevertheless, these different therapeutic strategies are still limited in both number and efficacy. Appropriate, clinically relevant animal models are essential for understanding the pathogenesis of impaired wound healing and for developing and establishing improved treatment approaches [14]. 

Several preclinical models for impaired wound healing can be found in the literature [15]: chronic pressure ulcer models [16,17,18,19], models addressing metabolic pathologies (e.g., diabetic wound healing models [20]), immunosuppressive strategies (glucocorticoids, radiation [21] or chemotherapeutics [22]), blood vessel ligation flap models [9], rat dorsal flap models [23,24], rabbit ear excision models [25], porcine ischemic wound models [26], and hindlimb models [27]. However, numerous human morbidities that negatively impact the healing process need to be considered when developing impaired wound healing models [14]: disturbances in perfusion, venous insufficiency, neuropathy, wound infection, age, immune system disease, obesity, metabolic diseases, and wound location all impair the physiologic healing process. However, impaired wound healing is uncommon in animals and none of the existing models is efficient in mimicking all relevant negative factors. Furthermore, investigations of treatment and application strategies are difficult in iatrogenic wounds, especially in small animals, and can lead to high dropout rates [15]. 

To address these clinical and economic issues of chronic wounds, we previously developed a new impaired-wound healing model in the rat [28]. For this we modified an already established rodent angiogenesis flap model [9]. We created one ischemic and one non-ischemic wound within the same animal on an abdominal flap by ligating a unilateral inferior epigastric bundle including artery, vein, and nerve. Two circular wounds were created in the abdominal skin flap, one in the ischemic and one in the non-ischemic area. The non-ischemic wound served as an internal control. Wounds in the ischemic region were of significantly larger size after seven days and had less angiogenesis in histologic analyses. The non-ischemic areas produced normal, well-perfused granulation tissue and showed proper progress in wound healing. This new model includes a decrease in oxygenation, venous insufficiency, and neuropathy and can easily be extended to animals suffering from immune system disturbances, obesity, or systemic metabolic diseases. Additionally, no control animals are needed as the contralateral non-ischemic side represents a more favorable internal control. This model has been used to test biophysical [29] and topical growth factor applications [30]. The growth factor used was a modified recombinant VEGF_164_ covalently bound to fibrin for slow release called TG-VEGF_164_-(where TG stands for being bound via TransGlutaminase). While this approach was highly successful, it requires the production of a specialized non-natural molecule with all the difficulties such a regulatory process entails. However, it is known that VEGF_165_ has a natural binding site to fibrin [31], which we have successfully used in one of our previous flap experiments [9]. Therefore, in this study we used our recently developed model to investigate the efficacy/dose dependency of naturally occurring VEGF_165_ in fibrin sealant in parallel ischemic and non-ischemic excision wounds [28].

Vascular endothelial growth factor (VEGF) stimulates wound healing by different mechanisms, including collagen deposition, angiogenesis, and epithelialization. The proangiogenic function comprises an induction of vessel permeability, endothelial cell migration, and endothelial cell proliferation [32,33]. Formation of granulation tissue in wound healing depends on angiogenesis and vessel infiltration to provide nutrition, mediators, and removal of metabolites. In normal acute wound healing, VEGF is upregulated between day 3 and day 7 (during the proliferation phase) [33]. VEGF reduction occurs after the wound granulates, angiogenesis ends, and the newly formed vessels decrease [33]. Clinical studies have investigated he proangiogenic effects of VEGF to promote wound healing in impaired wounds, including those occurring because of diabetes or arterial occlusive diseases [33]. Insufficient vascularization leads to delayed wound closure, reduction in epithelialization, and deficiencies in granulation tissue formation [34]. Interestingly, contradictory reports have been published about the correlation between angiogenesis and wound repair in healthy animals, with some studies describing no or only mild effects of inhibited angiogenesis on wound repair [35,36] and others describing significant impairment in wound healing [34]. 

Fibrin sealant is known to stimulate wound healing by activating angiogenesis, collagen synthesis, wound contraction, and epithelialization [37,38]; fibrin sealant is frequently used as a biomatrix for growth factor release. Different VEGF isoforms using fibrin sealant as a carrier for release were investigated for their ability to enhance angiogenesis. VEGF_165_ is mainly bound to ECM and the cell surface and has both heparin binding and direct binding sites for fibrin [31,39], leading to slow and controlled protease-mediated release [9,40]. In contrast, VEGF_121_ is a non-heparin and non-fibrin binding isoform [39,40] that leads to an uncontrolled burst of growth factor when mixed in fibrin [41]. An engineered variant of VEGF_121_, α2PI_1–8_-VEGF_121_, induces angiogenesis more potently than native VEGF_121_, due to its slow cell-mediated release [41]. Sacchi et al. investigated fibrin- α2-PI1–8-VEGF_164_, a murine VEGF isoform covalently crosslinked to fibrin with dose-dependent VEGF release. Optimized VEGF delivery led to proper angiogenesis in both an angiogenesis model and our delayed wound healing model [30].

In this study we aimed to investigate innovative, easily producible, and urgently required therapeutic approaches for chronic wounds. Therefore, we treated ischemic and non-ischemic wounds in our previously developed impaired wound healing model [28] with different doses of unmodified rhVEGF_165_ in a fibrin matrix or with the vehicle fibrin matrix only. Animals without therapy represented the control group. We investigated both the wound healing response and its dose dependency, paying special attention to the ischemic areas. Differences in wound healing between the therapeutic agents on the one hand, and the ischemic and the internal control wounds on the other hand, were then analyzed. As such, the current study should show the utility of our recently developed wound healing model for angiogenic therapeutic interventions.

## 2. Materials and Methods

### 2.1. Animal Model

The development of the animal model and analysis methods were recently published [29]. The local ethical committee on animal experiments in Vienna, Austria, approved all experimental in vivo procedures prior to study initiation. Animals were caged in pairs, with water ad libitum and free dietary access. Male Sprague Dawley rats (300–350 g) were initially anesthetized with isoflurane (2.5 Vol%). Anesthesia was maintained by intraperitoneal injection of ketamine hydrochloride (110 mg/kg) (Pharmacia & Upjohn, Erlangen, Germany) and xylazine (12 mg/kg) (Bayer, Leverkusen, Germany). The animal’s abdominal region was shaved and depilated. Surgeries were done under aseptic conditions. Butorphanol (1.25 mg/kg) and meloxicam (0.15 mg/kg) were injected subcutaneously (s.c.) as analgesic treatment on the day of surgery and on three consecutive post-op days. Peri-operatively, 10 mL Ringer solution was applied s.c. for fluid resuscitation. On day 7 post-op, the animals were euthanized by an intracardial overdose of barbiturate (Thiopental^®^, Abbott Laboratories, Chicago, IL, USA) 150 mg/kg BW). The surgical procedure was initiated with a cranial horizontal incision and the harvesting of an epifascial flap. The lateral flap borders were cut through and the unilateral inferior epigastric neurovascular bundle, including artery, vein and nerve, was ligated and excised to exclude spontaneous re-anastomosis. The caudal border was left intact, and the flap was sutured back in its original position using the interrupted suture technique. Thereafter, standardized circular wounds (diameter of 1.5 cm) were excised within the ischemic and non-ischemic areas, comprising skin and panniculus carnosus muscle, while leaving the underlying fascia intact. After therapy application, all excision wounds were covered with a transparent film dressing and fixed with a second dressing. Bandages were changed on days 1 and 3 after wound creation or replaced when necessary. 

### 2.2. Study Groups

Different treatments were applied to the wounds: a fibrin biomatrix (fibrin sealant) without additional factors as well as three different doses of recombinant human vascular endothelial growth factor 165 (rhVEGF_165)_ protein delivered locally from a fibrin biomatrix. The fibrin matrix used in the VEGF groups and the fibrin-only group were identical (see below for dosing and preparation). VEGF is a potent regulator of physiologic and pathologic angiogenesis. Studies applying VEGF-A as a pro-angiogenic factor (recombinant protein and gene therapy) in rodent models have already been performed in our lab [9,42]. Fibrin sealant is known to stimulate angiogenesis and wound healing to a certain degree [37,38]. One group without treatment comprised the control group. Additionally, the contralateral non-ischemic wound area served as an internal control in all study groups. Both the ischemic and non-ischemic sites of a single animal were always treated with the same therapy (Figure 1).

#### 2.2.1. Fibrin Biomatrix Group

A two-component fibrin sealant (FS) (Tisseel VH/SD, Baxter, Vienna, Austria) was used in this study. The FS was prepared according to the manufacturer’s instructions. The excision sites were treated with 0.2 mL fibrin sealant per wound, consisting of 0.1 mL fibrinogen solution and 0.1 mL diluted thrombin solution (4 IU/mL). For application, the fibrinogen and thrombin components were reconstituted with aprotinin (3000 KIU/mL) and calcium chloride (40 µmol/mL), respectively, and aspirated into a two-syringe applicator (Duplojects, Baxter, Vienna, Austria). Both components were mixed at the tip of the applicator and then 0.2 mL FS was directly applied to each wound (*n* = 9). 

#### 2.2.2. rhVEGF_165_ Group

For the VEGF protein treatment, the excision sites were covered with different amounts of rh VEGF_165_ (Pepro Tech, Rocky Hill, NJ, USA) in a fibrin matrix. The rhVEGF_165_ was added to the fibrinogen solution and mixed (1 + 1) with the thrombin solution (4 IU/mL) via the two-syringe applicator upon application equivalent to the vehicle fibrin biomatrix group. We compared 10 ng (*n* = 6), 50 ng (*n* = 6) and 100 ng (*n* = 8) rhVEGF_165_ per wound (these correspond to 100 ng, 500 ng, and 1000 ng rhVEGF_165_/mL fibrinogen per wound). 

#### 2.2.3. Control Group

Excision sites were left untreated and empty (*n* = 7).

### 2.3. Analyses

#### 2.3.1. Laser Doppler Imaging

Flap perfusion was measured using laser Doppler imaging (Moor Instruments Ltd., Devon, UK). Rat abdominal skin was scanned by a low-intensity laser light beam. The change in reflected laser frequency by moving blood cells was measured and a two-dimensional image of tissue perfusion was created. Tissue perfusion was reflected by a color-coded image: blue areas represented low perfusion and green, yellow, and red areas represented high perfusion. Scans were performed preoperatively, postoperatively, and on days 1, 3, and 7. The abdominal skin was divided into three areas representing tissue perfusion: a non-ischemic area with an intact epigastric bundle, an ischemic area with a ligated epigastric bundle, and a borderline zone in between. Non-ischemic site scans were set at 100%; ischemic site scans were referred to the non-ischemic site and expressed as a percentage. LDI scans were performed to both confirm ischemia postoperatively and track perfusion over the observation period.

#### 2.3.2. Planimetric Analysis

Planimetrical analysis was performed to measure wound size, follow wound closure, and compare wound healing progress between the ischemic and non-ischemic areas. Wound size was traced on a transparent acrylic sheet and photographed postoperatively (=day 0) and then at days 1, 3, and 7. Wound size was analyzed with planimetric software (Lucia G1, Version 4.8, Laboratory Imaging Ltd., Praha, Czech Republic). The total postoperative (=day 0) wound surface area was set at 100%; the wound sizes on days 1, 3, and 7 were compared to day 0 and expressed as a percentage. 

#### 2.3.3. Histological Analysis

Histological analysis was performed to reveal vessel growth, epithelialization, and invasion of immune cells in both the ischemic and non-ischemic areas. The area of ischemia was expected to contain low perfused tissue with few vessels but not extended necrosis. After animals were euthanized, the wounds were excised in toto, including the underlying muscle. The tissue was fixed in buffered 4% formaldehyde for 24 h and embedded in paraffin after a standard dehydration regimen. Sections were cut and stained with hematoxylin and eosin (H&E) or used for immunohistochemical procedures. 

#### 2.3.4. Immunohistochemical Analysis

Immunohistochemical staining of smooth muscle actin (SMA) was performed to visualize mature functional vessels in the wound area. The tissue was cut in 1.5 mm-thick sections and slides were heated to 60 °C for 30 min. Slides were then deparaffinized by washing with xylene twice for 10 min, with ethanol in increasing concentrations (50%, 70%, 99% and 100%) for 5 min each, and finally with phosphate-buffered saline (PBS) buffer for 2 min. Pronase type XXIV (Sigma Aldrich, St. Louis, MO, USA) 0.1% in PBS buffer was used for pretreatment of slides for 10 min and endogenous peroxidase (1% H_2_O_2_ in methanol) at room temperature for 15 to 30 min. Primary SMA antibodies (mouse anti-human; Dako A/S, Glostrup, Denmark), diluted 1:2000 in antibody reagent solution (Dako A/S), were incubated for 60 min at room temperature. Secondary antibodies (HRP mouse (EVN); ChemMate Dako Envision, Glostrup, Denmark) were added and incubated for 30 min at room temperature. A 5 min reaction with detection substrate AEC chromogen (Dako A/S) was carried out and then stopped in PBS. The samples were counter-dyed with hemalaun and fixed on object slides with Aquatex (Merck, Darmstadt, Germany), and then blind evaluation was carried out. The object slide was photographed, and three areas of observation (wound center and both wound edges) were marked per sample. Red-stained vessels were counted within these three fields and mean values were calculated.

#### 2.3.5. Statistical Analysis

Prism version 9.01 software (GraphPad Software, Boston, MA, USA) and Bonferroni’s *t*-tests were used to compare data between the groups. Results were expressed as means ± standard deviation (SD). Statistical significance was indicated by *p*-values < 0.05.

## 3. Results

### 3.1. A Decrease in Flap Perfusion in the Ischemic Flap Area Was Assessed by Laser Doppler Imaging

Laser Doppler imaging was performed to measure flap perfusion before and after vessel ligation. Non-ischemic site scans were set at 100%; the ischemic site scans were referred to those of the non-ischemic site and expressed as a percentage. The expected decrease in perfusion in the ischemic flap area compared to the non-ischemic area after ligation was seen in all groups. 

Before ligation, flap perfusion in the corresponding later ischemic area was 104.9 ± 10.2% in the VEGF groups, 108.3 ± 8.4% in the sham group, and 98.6 ± 12.3% in the fibrin group compared to contralateral perfusion. After ligation, the ischemic area LDI signal was 61.6 ± 11% (VEGF), 69.9 ± 10.2% (fibrin), and 60.2 ± 5.9% (sham) compared to the non-ischemic area. Over 7 days insufficient perfusion persisted in all groups in the ischemic area: 61 ± 10% (VEGF), 65.1 ± 10% (fibrin) and 59.5 ± 12.6% (sham). Ischemia was evident in all groups over the required observation period (Figure 2).

### 3.2. Planimetrical Analysis Showed Differently Disturbed Wound Healing in the Treated Ischemic Areas

Planimetric analysis was performed to measure wound size postoperatively (=day 0) and on days 1, 3, and 7. The total postoperative wound surface area was set as 100% and the wound areas on day 7 were expressed as a percentage.

After 7 days, wound areas in all groups had decreased compared to day 0. However, wound healing in ischemic areas was slower than in vital areas, with significant differences in the sham (mean 33.9 ± 19% vs. 69.7 ± 23.7%; *p* < 0.01), VEGF100 (mean 30.7 ± 8.5% vs. 58 ± 20.4%; *p* < 0.005) and VEGF50 group (32.3 ± 12.5% vs. 51.4 ± 9.3%; *p* < 0.05). No significant differences between vital and ischemic wounds were evident in the fibrin group (mean 38.4 ± 21% vs. 29.5 ± 18.8%) or in the VEGF10 group (mean 48.7 ± 13.9 vs. 34.5 ± 12.9). Fibrin-treated wound healing occurred faster compared to the other groups—irrespective of tissue vitality—with major differences in ischemic wounds and with significant differences between the fibrin and sham groups (*p* < 0.05).

The lower-VEGF dosing (10 and 50 ng) showed a trend to smaller wound areas (=potentially better healing) after 7 days compared to 100 ng VEGF in the ischemic wounds (no significant differences) (Figure 3).

### 3.3. H&E Staining Shows Differences in the Constitution of Newly Formed Granulation Tissue 

In all groups, H&E staining revealed only a few vessels in the wound area and surrounding tissue in ischemic sections. Similarly, we found necrotic tissue with high invasion levels of inflammatory cells in all groups.

Vital areas generated granulation tissue that consisted of newly formed vessels and infiltrated inflammatory cells. Epithelial cell invasion started from the wound edges. Wounds in the vital area with different treatments showed similar progress in wound healing (Figure 4).

### 3.4. Detection of Neovascularization in the Wounds

Immunohistochemical analysis (SMA staining) was performed to visualize neovascularization in the wounds. Histologic slides were photographed and three areas (in the wound center and in each wound edge) were marked. Red-stained vessels within the three areas were counted, the numbers were added, and statistical analysis was performed.

Statistical analysis revealed no significant differences between ischemic and non-ischemic wounds in the sham (237 ± 116 vs. 472 ± 389), VEGF100 (426 ± 322 vs. 722.4 ± 590.2), VEGF50 (897 ± 298 vs. 1091 ± 322.5), and VEGF10 (657 ± 404 vs. 767.2 ± 348.9) groups, but significantly less vascularization in the ischemic area compared to the non-ischemic area in the fibrin-treated group (415 ± 105 vs. 718 ± 276; *p* < 0.05).

Vessel density was compared between the different treatment groups. The ischemic regions displayed significantly more vascularization in the fibrin group compared to the sham group (*p* < 0.01), but the VEGF10 and VEGF50 group were associated with the highest vessel numbers, with significant differences compared to the sham (*p* < 0.0001), fibrin (*p* < 0.05), and VEGF100 groups (*p* < 0.05). No significant differences were found between the fibrin and VEGF100 groups or between the VEGF10 and VEGF50 groups.

In the vital area, analysis revealed significantly more vessels in the VEGF50 group compared to the sham (*p* < 0.01) and fibrin (*p* < 0.05) groups but no significant differences between the other groups (Figure 5 and Figure 6).

## 4. Discussion

The urgent demand for innovative and easy therapeutic approaches to chronic wounds remains a serious clinical problem. Using our previously developed rodent model [28], in this study we treated ischemic and non-ischemic wounds with different therapeutic agents to investigate wound healing response in differently perfused wound environments. For this purpose, fibrin only and three different doses of rhVEF_165_ delivered by a fibrin matrix were applied to wounds. We administered the same treatment to both wound regions to avoid local migration or systemic effects (and consequent effects on the contralateral wound). The excisional wounds provide a circular area of wound surface for easy application of therapeutic agents in different constitutions or via different application methods, including injection, flat-surface application of sealants and powders, or the application of meshes. We detected an advantage over other ligation models (e.g., rabbit ear models), which are limited due to the lack of a planar wound surface. Furthermore, our model allows therapeutic approaches that have been clinically established but are not yet fully mechanistically understood, like shockwave therapy [7,29] to be studied to investigate associated pathways in healthy and disturbed wound healing conditions. However, one of the biggest strengths of our model is the possibility to test these treatment approaches in both an ischemic wound area and a regularly perfused, contralateral, internal control wound. Thus, differences in the wound healing process can be elucidated in one animal, excluding a certain variability evident in different individuals. Both noninvasive laser Doppler imaging and computed planimetrical analysis were performed to reveal ischemia and follow-up wound size over seven days. LDI found decreased perfusion in the ischemic area compared to the non-ischemic area in all groups. 

Patil et al. modified a porcine ischemic wound model. They created bipedicle flaps with excisional wounds in ischemic and non-ischemic areas and treated the wounds with two different scaffolds. They found tissue infiltration in both the ischemic and non-ischemic wounds after 10 days, but markedly lower perfusion, tissue infiltration, and M2 macrophage activity in the ischemic wounds. Further, they reported significant differences in tissue infiltration and perfusion between the two scaffolds in the ischemic but not non-ischemic wounds. Patil et al. suggested that biomaterial testing in ischemic wounds provides more sensitivity compared to non-ischemic wounds [26]. Similar to their findings, we detected significantly higher vessel density after VEGF therapy application in ischemic but not non-ischemic wounds. We assume that the induced chronic wounds respond well to proangiogenic therapy, an effect we did not detect in the non-ischemic wounds. It is known that the mitogenic, chemotactic, and permeability effects of VEGF promote angiogenesis in chronic wounds in both arterial occlusive diseases and diabetes [33]. These effects were likely responsible for stimulating neovascularization in the VEGF- treated ischemic wounds. 

Like Patil et al., we induced acute ischemia by the ligation of the epigastric bundle, which does not exactly mimic the chronic ischemic situation that appears in pressure or diabetic ulcers. Nevertheless, we were able to induce an ischemic environment that led to an impaired wound with different responses to therapeutic agents. In the future, the model can be applied in animals with metabolic disease, like diabetic or adipose rats, which is a further crucial advantage of our impaired wound healing model.

Although it is known that VEGF promotes healing in chronic wounds [33], we did not observe more rapid wound closure in ischemic areas with treatment of 100 ng of rhVEGF_165_. We additionally compared the 100 ng rhVEGF_165_ with lower VEGF doses (50 and 10 ng per wound), as non-functional angiogenesis is suggested to be caused by over-expression of VEGF [43]. Both 50 and 10 ng rhVEGF_165_ per wound, delivered by the fibrin biomatrix, led to smaller wound areas after 7 days in the ischemic area compared to 100 ng rhVEGF_165_. We also detected significantly higher vessel numbers in the wounds treated with 10 and 50 ng rhVEGF_165_. High levels of VEGF are known to result in the development of leaky and immature vessels. Greenberg et al. described an aberrant vessel growth induced by insufficient pericyte stabilization using high VEGF concentrations through induction of VEGFR-2/PDGFR-β complexes [44]. These findings may explain our unexpected results. Interestingly, we detected the fastest wound closure in the ischemic wounds treated only with fibrin, although vessel counting revealed less wound vascularization compared to all VEGF groups. Fibrin is itself known to promote wound healing by activating angiogenesis, collagen synthesis, wound contraction, and epithelialization [38]. Possibly, fibrin-induced angiogenesis results in vessels of higher quality compared to that induced by VEGF treatment. A review by Abbade et al. emphasized the healing properties of fibrin sealant in chronic venous ulcers demonstrated in various studies [37]. Eming et al. investigated a possible link between low VEGF_165_ concentration in chronic wounds and impaired wound healing, but they detected upregulated VEGF-A mRNA and protein expression in chronic wounds comparable to that in normally healing wounds. Thus, they focused on the degradation pathway of VEGF_165_ protein in chronic wounds; they found that elevated proteolytic processes targeting VEGF_165_ led to increased protein degradation compared to that in normally healing wounds [32]. As we discussed above, we induced acute ischemia with consecutive impaired wound healing in our recently established model. We did not investigate possible VEGF proteolysis in our study setup, but we detected decreased vessel infiltration in the sham group compared to the VEGF-treated groups. 

Overall, we detected the most rapid wound closure in the fibrin-only group despite less vascularization in the wound area compared to all VEGF concentration groups. This was possibly caused by aberrant or insufficient vessel growth in the VEGF treated wounds, or, less likely, by VEGF degradation within an ischemic chronic wound milieu.

## 5. Conclusions

In this study we investigated the dose-dependent effects of naturally occurring rhVEGF_165_ in a fibrin sealant matrix in ischemic and non-ischemic excision wounds using our previously developed impaired wound healing model. Different rhVEGF_165_ dosing led to differences in angiogenesis and wound healing in the impaired wounds. In lower doses rhVEGF_165_ promoted neovascularization within the ischemic wound areas but surprisingly did not increase wound closure compared to the fibrin-only group. However, both the fibrin and rhVEGF10 groups induced the most rapid wound closure in the ischemic wounds and showed the smallest differences in wound size (compared to vital wound sites) as well. Overall, we were able to investigate the benefits of VEGF treatment in insufficiently perfused wounds and demonstrate the utility of our previously established impaired wound healing model. 

## Figures and Tables

**Figure 1 biomedicines-11-01043-f001:**
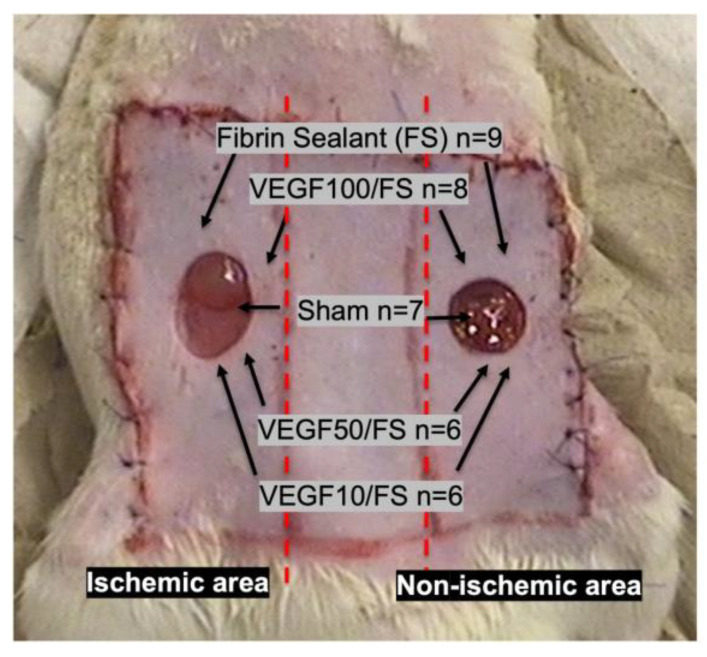
Animal model and group allocation. We created one ischemic and one non-ischemic wound on an abdominal flap within the same animal by ligating a unilateral inferior epigastric bundle. Both wounds were treated with the same therapy to prevent systemic interaction between different treatments. FS = fibrin sealant, VEGF = vascular endothelial growth factor, VEGF 100 FS = 100 ng VEGF in fibrin sealant, VEGF 50 = 50 ng VEGF in fibrin sealant, VEGF 10 = 10 ng VEGF in fibrin sealant. Sham = without treatment. Red interrupted line indicates ischemic and non-ischemic area.

**Figure 2 biomedicines-11-01043-f002:**
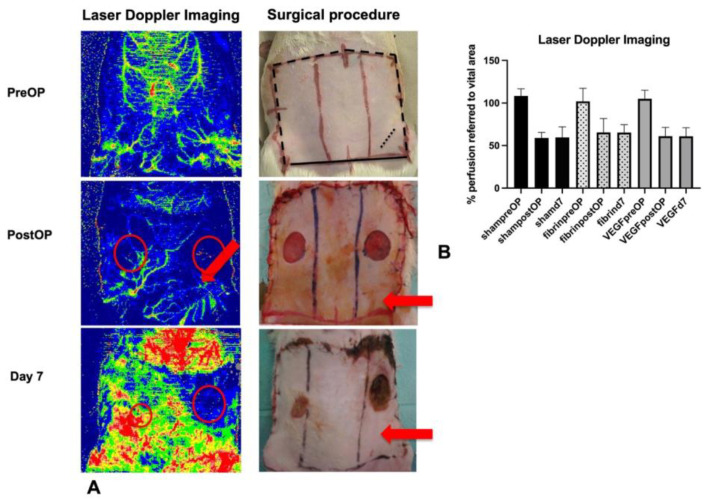
Laser Doppler imaging (LDI) and surgical situs. LDI scans were performed to both confirm ischemia postoperatively and track perfusion over the observation period. (**A**): Laser Doppler images demonstrating tissue perfusion and corresponding pictures of the surgical situs at different time points: green, yellow, and red areas indicate high perfusion; regions colored blue or dark blue reflect low perfusion. Red circles mark the location of the excisional wounds; red arrows mark the ligation side. Compromised perfusion of the ligated wound area was achieved over 7 days. (**B**): LDI analysis: non-ischemic site scans were set at 100%, the ischemic site scans were referred to those of the non-ischemic site and expressed as a percentage. The graph bars depict the tissue perfusion in the ischemic area in differently treated groups. The tissue perfusion in all groups decreased immediately after ligation, and impaired perfusion persisted during the 7-day observation period. PreOP = before ligation, PostOP = after ligation, day 7 = after 7 days. Sham = without treatment, fibrin = fibrin treatment, VEGF = VEGF 100 (vascular endothelial growth factor) treatment.

**Figure 3 biomedicines-11-01043-f003:**
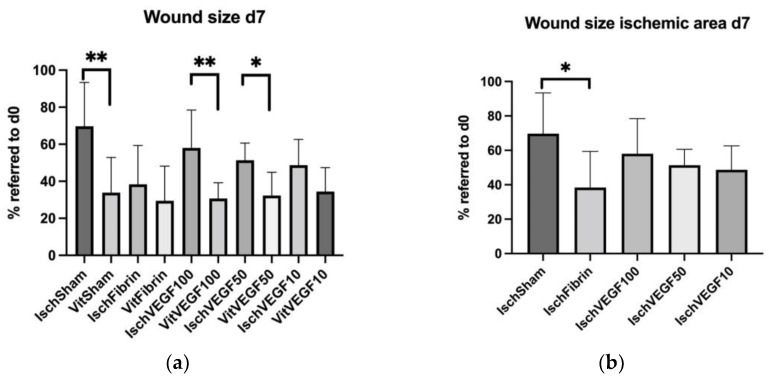
(**a**) Planimetric analysis of ischemic and non-ischemic wounds on day 7, (**b**) Planimetric analysis of different treatments in the ischemic area on day 7. Colored graphs show the wound size of the differently treated groups in differently perfused areas on day 7. The postoperative wound surface area was set at 100% and the wound areas on day 7 were expressed as percentages. (**a**) Wound size comparison between ischemic and non-ischemic areas. Wound closure in the ischemic area occurred more slowly compared to the non-ischemic wounds in all groups with significant differences in the sham, VEGF100 and VEGF50 groups. (**b**) Wound size in the ischemic area compared between the different treatments. The fibrin- treated wounds showed the smallest wound size after 7 days, with a significant difference compared to the sham group. (mean ± SD, *n =* 6–9, * *p* = < 0.05, ** *p* < 0.01) Sham = no treatment, fibrin = fibrin treatment, VEGF= vascular endothelial growth factor, VEGF100 = 100 ng VEGF treatment, VEGF50 = 50 ng VEGF treatment, VEGF10 = 10 ng treatment; Isch = ischemic, Vit = vital (non-ischemic).

**Figure 4 biomedicines-11-01043-f004:**
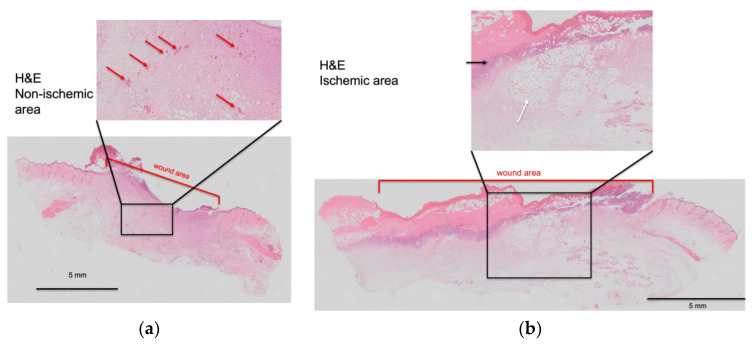
Representative H&E images of fibrin-treated wounds. Representative histologic images of fibrin-treated ischemic and non-ischemic wound areas. The top images represent the magnified image of the area enclosed in the black rectangle of the image below. Red bars depict wound area; red arrows indicate vascular structures; white arrow indicates vacuolar degeneration; black arrow indicates inflammatory cells. (**a**) Non-ischemic area: The granulation tissue shows a mild inflammatory response in the non-ischemic wounds with vascular structures (arrows) and epithelialization starting from wound edges. (**b**) Ischemic area: Granulation tissue growth and angiogenic response was impaired in the ischemic wounds. Histologic images show degenerative vacuoles as a sign of tissue necrosis in the ischemic area.

**Figure 5 biomedicines-11-01043-f005:**
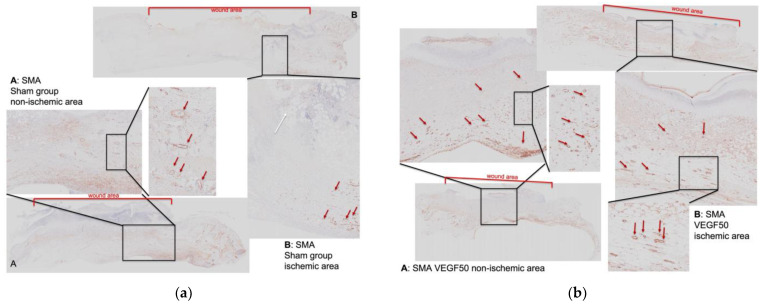
Immunohistochemical staining demonstrates mature vessel ingrowth. Immunohistochemical staining with Smooth Muscle Actin (SMA) antibodies was performed to demonstrate mature vessel infiltration within the granulation tissue. The top images represent the magnified image of the area enclosed in black rectangle of the image below. Bars depict wound area; red arrows indicate vascular structures; white arrows indicate degenerative vacuoles. (**a**) Sham SMA (*n* = 7) **A**: Non-ischemic area: immunohistochemical staining shows regular vessel infiltration within the wound. **B**: Ischemic area: rare vessel infiltration occurred within the granulation tissue with less vessel sprouting from the adjacent tissue. Degenerative vacuoles demonstrate necrosis in the wound area. (**b**) VEGF50 SMA (*n* = 6) **A**: Non-ischemic area: immunohistochemical staining reveals increased vessel infiltration within the wound. **B**: Ischemic area: extensive vessel infiltration occurred within the granulation tissue and the adjacent tissue. Some degenerative vacuoles are present in the granulation tissue as well. SMA = Smooth Muscle Actin, Sham = non-treated wounds, VEGF50 = 50 ng vascular endothelial growth factor.

**Figure 6 biomedicines-11-01043-f006:**
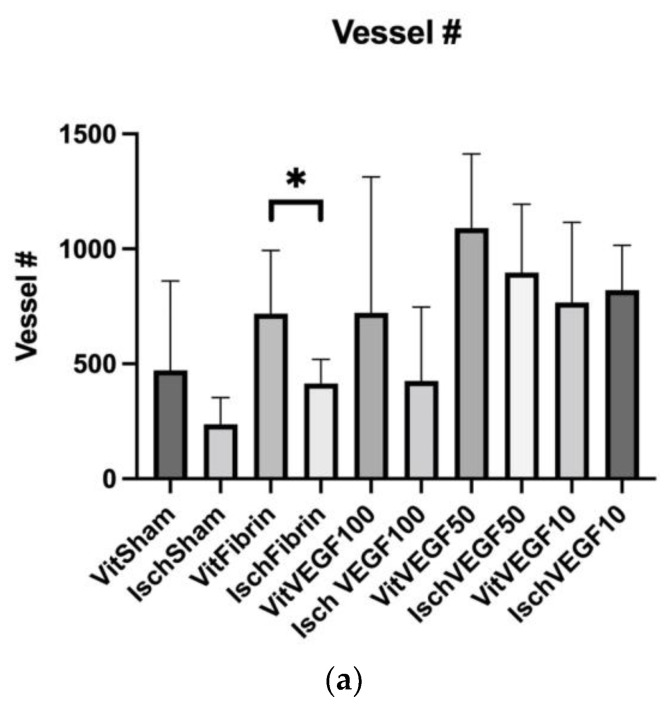
Quantification of neovascularization and comparison between non-ischemic and ischemic area (**a**); between treatments in the non-ischemic area (**b**) and ischemic area (**c**). The colored graph bars depict the vessels numbers in differently treated groups and differently perfused areas. (**a**): Comparison between differently perfused wound areas (ischemic vs. non-ischemic). Vessel counting revealed a decrease in vascularization in ischemic wounds compared to non-ischemic wounds in all groups, with significant differences in the fibrin-only treated wounds. (**b**): Comparison of vessel numbers between the different therapies in the non-ischemic (vital) area: Vessel counting showed significantly higher vessel numbers in the VEGF50 treated group, with significant differences compared to the fibrin and sham groups. (**c**): Comparison of vessel numbers between the different therapies in the ischemic area: we found the highest vascularization in the VEGF50 and VEGF10 groups, with significantly more vessels than in the sham, fibrin and VEGF100 groups. The sham group demonstrated the lowest vessel numbers. (mean ± SD, *n* = 4–9, * *p* < 0.05, *** *p* < 0.005). Sham = without treatment, fibrin = fibrin treatment, VEGF = vascular endothelial growth factor, VEGF100 = 100 ng VEGF treatment, VEGF50 = 50 ng VEGF treatment, VEGF10 = 10 ng treatment; Vessel# = vessel numbers, Isch = ischemic, Vit = vital (non-ischemic).

## Data Availability

The authors confirm that all data supporting the findings of this study are available within the article. Raw data supporting the findings of this study are available from the corresponding author (R.M.) on request.

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
