# Peer review of "Ischemia Impaired Wound Healing Model in the Rat—Demonstrating Its Ability to Test Proangiogenic Factors"

_biomedicines, 2023, doi:10.3390/biomedicines11041043_

Round 1

Reviewer 1 Report

In this manuscript, Hofmann, Mittermayr, et al. describe the effects of fibrin and VEGF on wound healing of the normal (vital) and ischemic skin tissues using a mouse model system developed by the authors’ group. I find the topic quite interesting and suitable for Biomedicines. I, however, have several points that I would like to ask the authors to consider before publication.

Major points:

1)    The data presented in section 3.4 are hard to grasp and evaluate. It is probably because the authors describe a lot of numbers in the text. It would be better to use graph(s) or table(s), including p-values for all possible combinations, to facilitate readers trying to evaluate data by themselves.

2)    It was not clear to me whether the “fibrin sealant” used in the “fibrin biomatrix group” and the “fibrin matrix” used in the “rhVEGF165 group” are equivalent or not. In other words, is fibrin a kind of vehicle control for VEGF-treatment, or VEGF-treatment and fibrin-treatment should be considered as two independent treatments?

3)    Lines 275-277, “Interestingly, ---”: Are the authors making conclusion based on data with no significant difference?

4)    In section 3.3, I got the impression that the authors make several important points about the number and distribution of blood vessels just showing a few micrographs (without quantitative data). I was not sure whether these statements are somehow related to the data presented in section 3.4 or not.  

5)    Special care should be taken to clarify that this paper and reference No. 9 (and any other papers) are not duplicate publication. I understand that the authors are conscientious about this issue, but the difference between the two studies are not so evident from the text. It would be better to more emphasize what was the technical improvement, if any, or why the present study was essential if the technique used was the same as previous one. If this study is to evaluate the utility of the previously developed technique, it is not appropriate to say “We established an ---”(line 21) in the conclusion section (item 4) in Abstract of this paper.  

Minor points:

6)    Line 12, “in the rat”: The preposition “from” instead of “in” should be more appropriate in this context.

7)    Line 15, “and fibrin alone”: The conjunction “or” instead of “and” should be more appropriate in this context.

8)    Line 17: Is the word “Results” missing after “(3)”?

9)    Line 59, “iatrogenic-induced wounds”: The phrase sounds redundant to me: why not “iatrogenic wounds”?

10) Line 60, “dropout rates”: What are the causes of dropout?

11) Line 74, “TG-VGF164”: What does “TG” mean?

12) Lines 111-118: Last two paragraphs in Introduction seem to be redundant. Besides, this is the place where the authors could emphasize the difference/novelty of this study (see my comment #5).

13) In Materials and Methods, each section usually consists of one paragraph (no indentation).

14) Line 292, “revealed fewer vessels”: Fewer than what?

15) Line 296, “evenly regular numbers”: The phrase does not make sense to me; it sounds like a mathematical term.

16) “Following the urgent requirement/demand” (lines 111 and 340): The word “following” sounds a little out of place to me in these contexts.

Reviewer 2 Report

Since chronic wounds remain a serious clinical problem with insufficient therapeutic approaches the authors investigated the dose dependency of rhVEGF165 in fibrin sealant in ischemic and non-ischemic excision wounds in our recently developed impaired-wound healing model.

Methods: An abdominal flap was harvested in the rat with unilateral ligation of the 12 epigastric bundle and consequent unilateral flap ischemia. Two excisional wounds were set in the ischemic and non-ischemic area. Wounds were treated with three different rhVEGF165 doses (10, 50 14 and 100ng) mixed with fibrin and fibrin alone. Control animals received no therapy. Laser Doppler 15 Imaging (LDI) and immunohistochemistry were performed to verify ischemia and angiogenesis. Wound size was monitored with computed planimetric analysis. Results:a)  LDI revealed insufficient tissue perfusion in all groups. b) Planimetric analysis showed slower wound healing in the ischemic area in 18 all groups. c) Wound healing was fastest with fibrin treatment- irrespective of tissue vitality. d) Lower dose VEGF (10 and 50ng) led to faster wound healing compared to high dose VEGF. Immunohistochemistry showed highest vessel numbers in low dose VEGF groups.

Conclusions: The authors conclused to have established an impaired-wound healing model to investigate the effect of differently dosed rhVEGF165 and fibrin sealant with special attention on ischemic wounds.

Mayor concerns are the followings.

a) The authors investigates the response of vessels. How about the response of cellular infiltrate ?

b) Have the authors checked the behavior of fibroblasts?

c) The references abot chronic wounds are very obsolete. Is necessary found some refs of 2020-2022

Round 2

Reviewer 2 Report

The authors have answered correctly to my answers